# Knowledge, Beliefs, Dietary, and Lifestyle Practices Related to Bone Health among Middle-Aged and Elderly Chinese in Klang Valley, Malaysia

**DOI:** 10.3390/ijerph16101787

**Published:** 2019-05-20

**Authors:** Chin Yi Chan, Shaanthana Subramaniam, Kok-Yong Chin, Soelaiman Ima-Nirwana, Norliza Muhammad, Ahmad Fairus, Abdul Manap Mohd Rizal, Pei Yuan Ng, Jamil Nor Aini, Noorazah Abd Aziz, Norazlina Mohamed

**Affiliations:** 1Department of Pharmacology, Universiti Kebangsaan Malaysia Medical Centre, Cheras 56000, Malaysia; chanchinyi94@gmail.com (C.Y.C.); shaanthana_bks@hotmail.com (S.S.); chinkokyong@ppukm.ukm.edu.my (K.-Y.C.); imasoel@ppukm.ukm.edu.my (S.I.-N.); norliza_ssp@ppukm.ukm.edu.my (N.M.); 2Department of Anatomy, Universiti Kebangsaan Malaysia Medical Centre, Cheras 56000, Malaysia; apai.kie@gmail.com; 3Department of Community Health, Universiti Kebangsaan Malaysia Medical Centre, Cheras 56000, Malaysia; mrizal@ppukm.ukm.edu.my; 4Faculty of Pharmacy, Universiti Kebangsaan Malaysia Kuala Lumpur Campus, Jalan Raja Muda Abdul Aziz, Kuala Lumpur 50300, Malaysia; pyng@ukm.edu.my; 5Faculty of Health Science, Universiti Kebangsaan Malaysia Kuala Lumpur Campus, Jalan Raja Muda Abdul Aziz, Kuala Lumpur 50300, Malaysia; ainijamil@ukm.edu.my; 6Department of Family Medicine, Universiti Kebangsaan Malaysia Medical Centre, Cheras 56000, Malaysia; azah@ppukm.ukm.edu.my

**Keywords:** knowledge, attitude, perception, behaviors, diet, bone density, dual-energy X-ray absorptiometry

## Abstract

Osteoporosis is a growing health problem in Asian countries with a rapidly expanding aging population. Adequate knowledge and positive health beliefs regarding osteoporosis will encourage individuals to adopt measures to protect bone health. This study aimed to investigate the association between knowledge, beliefs, and practices regarding osteoporosis and bone health among Malaysians Chinese aged 40 years and above. A cross-sectional study was conducted among 367 Malaysians Chinese (182 men, 185 women) aged ≥ 40 years in Klang Valley, Malaysia. They completed a questionnaire on knowledge, beliefs, and practices of osteoporosis and underwent bone mineral density scan using a dual-energy X-ray absorptiometry device. The subjects showed moderate knowledge and high level of health beliefs regarding osteoporosis, but poor osteoprotective practices. Osteoporosis knowledge and beliefs were significantly different based on subjects’ demographic characteristics *(p* < 0.05). Additionally, osteoporosis knowledge was correlated positively with beliefs, coffee or tea intake (*p* < 0.05) but not with other lifestyle practices. Osteoporosis health beliefs was associated positively with physical activity, dairy and calcium intake (*p* < 0.05). However, bone health was not associated with knowledge, beliefs and practices regarding osteoporosis (*p* > 0.05). The present findings highlight the need of formulating osteoporosis prevention program targeting Malaysians Chinese, especially men, to improve their knowledge, health beliefs, and practice.

## 1. Introduction

Osteoporosis is a silent metabolic bone disease that can potentially cause fragility fractures, disability, and increased mortality due to weakened bone strength and reduction in bone mass [1]. Worldwide, one out of three women and one out of five men will experience an osteoporotic fracture during their lifespan after the age of 50 years [2]. The population of people over 50 years old in Asia is estimated to rise from 5.3 million in 2013 to 13.9 million in 2050 [3]. Correspondingly, by 2050, more than 50% of all osteoporotic fractures will occur in Asia [4]. The total burden of osteoporosis in Southeast Asia accounted for 0.56% of the global burden of non-communicable diseases in 2000 [1] and it is expected to rise with the increased elderly population. In Malaysia, it is estimated that hip fracture incidence will increase by 3.55 folds from 2018 to 2050 [5]. Consequently, this will increase the economic burden to the patients, their families, and the country. A previous report estimated that the inpatient costs of hospitalization due to hip fractures in Malaysia in 1997 was 22 million ringgits (RM), and this does not include rehabilitation and nursing home costs [6]. Despite the lack of recent data, the cost is increasing with inflation. As the median monthly household income among Malaysians was only RM 4585 [7], this represents a significant healthcare burden which the patients cannot afford.

Osteoporosis is associated with modifiable risk factors, such as sedentary lifestyle and imbalanced diet, as well as non-modifiable risk factors, such as sex, aging, and family history [8]. Knowledge and health beliefs regarding osteoporosis are the foundation for making informed lifestyle choices towards better bone health. Perception and knowledge can motivate individuals to adopt recommended preventive health actions [9]. Several studies on knowledge, beliefs, and practices regarding osteoporosis among Malaysians have been conducted [10,11,12,13,14]. The populations surveyed included adults [10,13,14] and undergraduate students [11,12]. These studies found that most subjects showed moderate bone health knowledge [10,13] with differences among the ethnic groups [12] and sex [11,12,14]. Specifically, women were more knowledgeable than men [11,12,14], and the Chinese [12] subjects were more knowledgeable than other ethnic groups. Higher income and education were also associated with better osteoporosis awareness [14]. Moderate [13] to good [10] health beliefs regarding osteoporosis were noted in these studies. The osteoprotective practices among Malaysians were generally poor [10,13]. Their practices were associated with sex [11], whereby men were more physically active but had a higher intake of caffeine and carbonated beverages than women [11]. However, there were some limitations to these studies. Firstly, correlation analysis between knowledge, beliefs, and practices regarding osteoporosis with bone health was not carried out [10,11,12,13,14]. Moreover, some studies did not include middle-aged and elderly populations, who are at risk for osteoporosis [11,12]. Hence, the barriers for the middle-aged and elderly population in attaining optimal bone health are not completely understood.

Critical understanding of individual knowledge and health beliefs regarding osteoporosis is necessary for planning osteoporosis prevention and lifestyle intervention program [15]. These domains are influenced by demography, cultural, and socio-economic features of the population studied [16]. The Malaysian Chinese population was the focus of this study because previous reports showed that they had lower bone mineral density and higher fracture incidence compared to the other ethnicities [17,18]. Hence, information about the knowledge, health beliefs, and practices on bone health among Malaysian Chinese is important to design osteoporosis prevention programs for them. This study aimed to evaluate the association between knowledge, beliefs, and practices regarding osteoporosis and bone health among middle-aged and elderly Chinese living in Klang Valley, Malaysia. This region, which is located in Central Malaysia, is highly urbanized and 43.3% of the population is Chinese [19].

## 2. Materials and Methods

This cross-sectional study was conducted from April 2018 to January 2019 in Klang Valley, Malaysia. The study protocol had been reviewed and approved by the Ethics Committee of Universiti Kebangsaan Malaysia Medical Centre (approval code: UKM PPI/111/8/JEP 2017-721). A total of 367 Malaysian Chinese aged 40 years and above were recruited using the quota sampling technique. Invitation with specific inclusion and exclusion criteria was sent to community centers in Klang Valley and advertised in local newspapers and radio stations. Potential participants were interviewed over the phone to ensure their eligibility. Subjects eligible to participate in this study were invited to a face-to-face interview session. During this session, they were informed of the project details and written consent was taken if they agreed to participate. They also answered questionnaires on their demography, knowledge, beliefs, and practices during this session. Subjects were given a date for body anthropometry and bone health measurements after completing the interview. 

Subjects previously diagnosed with osteoporosis, metabolic bone diseases (Paget’s disease, osteogenesis imperfecta, osteomalacia, and rickets), hypo/hyperparathyroidism, hypo/hypercalcaemia, hypo/hyperthyroidism, and/or were receiving pharmacological treatment for osteoporosis (bisphosphonates, teriparatide, denosumab, and strontium ranelate) or other treatments that could significantly impact bone metabolism (hormone-replacement therapy, sex hormone deprivation therapy, thiazide diuretics, anticonvulsants, antidepressants, glucocorticoids, and thyroid supplements) were excluded. Those having mobility problems, needing a walking aid, having metal implants at the calcaneus, hip, spine, or femoral neck, suffered a lower limb fracture 2 years prior to the screening date, or a low impact fracture after the age of 50 years, or could not complete the questionnaire or screening procedure were excluded as well. 

Subjects completed a questionnaire on their sociodemographic details, knowledge, beliefs, practices (diet, lifestyle, and physical activity) regarding osteoporosis during the interview. The age of the subjects was determined from the records on their identification card. Ethnicity, sex, menstrual status, age of menopause, and parity and presence of pre-existing medical conditions and medical treatments were self-declared. Subjects were classified into their respective age clusters (young, middle age, and elderly) according to the convention. Subjects aged 40–59 years were classified as “middle-age”, and this cut-off was supported by other researchers studying middle-aged population in Malaysia [20,21]. The elderly population was defined using the World Health Organization standard for developing countries as the population that had reached retirement age which is around 60 years in Malaysia [22]. Subjects’ occupation was categorized into manual or sedentary based on the amount of time they spent on walking or carrying heavy objects (manual) or sitting at the workplace or in a vehicle (sedentary). Subjects who were jobless, housewives, or retirees were categorized in the sedentary group. The subjects were also classified based on household income into the bottom 40% (B40, with household income < RM 7640) and the middle 40% (M40, with household income RM 7640–15,159) group according to data of the Malaysian census [23]. 

The knowledge questionnaire was modelled based on the Osteoporosis Prevention and Awareness Tool (OPAAT) [24]. The researchers involved in this study discussed and selected 12 questions from the original questionnaire based on current literature, in which six tested the general knowledge regarding osteoporosis and six tested knowledge on osteoporosis prevention. The subjects were required to answer “true”, “false”, and “don’t know” for each item. A correct answer was given 1 point and an incorrect answer or “don’t know” was given 0 point. The knowledge score was summed up and categorized into low (less than or equal to 50%), moderate (51% to 69%), and high (70% and above) [12,13]. The reliability coefficient obtained through test-retest in the pilot study (n = 30, 7 days apart) for the knowledge questionnaire was 0.739. 

Health beliefs of subjects regarding osteoporosis was determined using a 12-item questionnaire, modelled after the Osteoporosis Health Belief Scale (OHBS) [25]. The researchers of this study discussed and decided the items included in the questionnaire. The items covered subjects’ perceived susceptibility and seriousness to osteoporosis (items 1–3), benefits of exercise and calcium (items 4–5), barriers to exercise and calcium intake (items 6–9), and health motivation (items 10–12). Subjects scored each item using a Likert’s scale from 1 (strongly agree) to 5 (strongly disagree). The scores of negatively worded items were inversely coded by the researchers during analysis. The beliefs score was summed up and categorized into low (less than or equal to 50%), moderate (51% to 69%) and high (70% and above) [12,13]. The reliability coefficient obtained through test-retest in the pilot study (n = 30, 7 days apart) for the knowledge questionnaire was 0.731. 

In terms of practice, the subjects were requested to disclose their smoking behavior, intake of beverages, and intake of dairy products [26]. Subjects answered “yes/no” for the consumption of these products for the past seven days. If they answered “yes”, the type of products and the frequency of consumption (how many times per week) were investigated. The current smokers and subjects had ceased smoking for the past 12 months were combined as ‘ever-smokers’. The beverages and dairy products investigated included (1) coffee or tea (caffeine-containing beverages), (2) alcohol (beer, wine or spirits), (3) milk, (4) yoghurt, and (5) cheese. One unit of milk was defined as 200 mL whereas coffee/tea was defined as one standard coffee cup/tea cup. One unit of yoghurt and cheese was defined as a cup of yoghurt and a slice of cheese respectively. Alcohol unit was defined according to the recommendation by the National Health Service, UK [27]. A unit of alcoholic drink referred to a bottle of beer/cider, a glass of wine, or one portion of spirits/strong alcohol [28]. For beverages, subjects with an intake of less than 1 unit per week were defined as non-drinkers. Regular coffee/tea drinkers and dairy product consumers were defined as those who consume 1 unit of coffee/tea or dairy products for three to four days per week. For grouping according to the frequency of alcohol intake, subjects who consumed alcohol regularly (three to four days per week) or had stopped drinking for the past 12 months were combined as ‘ever-drinkers’. Those who never or rarely consume alcohol (one or fewer days per month) were combined as ‘non-drinkers’. As for calcium supplement, those who consumed at least one tablet of calcium supplement for three to four days per week were considered as regular users.

Physical activity status of the subjects was determined using the International Physical Activity Questionnaire (IPAQ), which is available online and free for use [29]. Briefly, subjects were required to recall the average amount of time spent in high-intensity activity, moderate-intensity activity, walking, and sitting/lying down (except sleeping) in a week. These were converted to the metabolic equivalent of task (MET) and summed up. Subjects were classified into inactive, minimally active, or HEPA (health-enhancing physical activity) based on the total MET score or other additional criteria [30]. The validity and reliability of this instrument have been tested in the Malaysian population [29]. 

Standing height of the subjects without shoes was measured using a stadiometer (Seca, Hamburg, Germany) and recorded to the nearest 1 cm. Body weight of the subjects with light clothing and without shoes was determined using a weighing scale (Tanita, Tokyo, Japan) and was recorded to the nearest 0.1 kg. Body mass index (BMI) of the subjects was calculated as per the convention: their body weight in kg divided by the square of their height in meters. Generally, for subjects below 65 years old, BMI <18.5 kg/m^2^ were underweight, 18.5–24.9 kg/m^2^ were normal, 25.0–29.9 kg/m^2^ were overweight, and >30.0 kg/m^2^ were obese [31]. For people above 65 years old, 22–27 kg/m^2^ were normal, >27 kg/m^2^ were overweight, and <22 kg/m^2^ were underweight [32]. The waistline of the subjects was measured by using a soft measuring tape and were recorded to the nearest 0.1 cm. The waist circumference was measured between the lowest rib margin and the iliac crest while subjects maintained a standing position. 

Bone mineral density (BMD) of the subjects at lumbar spine and femur of the non-dominant leg (femoral neck and total hip) were measured with the Hologic Discovery QDR Wi densitometer, DXA (Hologic, MA, USA). DXA is the gold standard in osteoporosis diagnosis. The DXA was calibrated daily with a phantom. The short term in-vivo coefficient of variation for the DXA machine was 1.8% and 1.2% for the lumbar spine and total hip, respectively. The body fat percentage, lean body mass, lumbar spine bone mineral density (average BMD of L1–L4), and hip bone mineral density were computed automatically by the DXA scanner. The T-score was generated by comparing the BMD values of the subjects with the reference values of the Singaporean population. According to the WHO guidelines, a T-score of ≤−2.5 indicates osteoporosis, between −2.5 and −1 indicates osteopenia, and more than >−1 indicates normal bone health status [33].

### Statistical Analysis 

Data were analyzed using the Statistical Package for Social Science Version 22 (IBM, Armonk, USA) software. Statistical significance was defined as *p* < 0.05. Normality of the data was tested using the Kolmogorov–Smirnov test. Descriptive summary measures (frequency distribution table) were expressed as mean (SD). Continuous variables and categorical data were expressed as percentages (for subject characteristics, knowledge, beliefs, and practices regarding osteoporosis). Knowledge and health beliefs regarding osteoporosis were summed up and categorized into low (less than or equal to 50%), moderate (51% to 69%), and high (70% and above) [12,13]. Independent T-test or one-way analysis of variance with post hoc analysis was used to determine the difference in sociodemographic characteristics and knowledge, beliefs and practices related to bone health. Pearson correlation was used to identify the interrelationship among knowledge, beliefs, and practices with subjects’ bone health status.

## 3. Results

### 3.1. Characteristics of Subjects 

A total of 367 subjects comprising of 182 men and 185 women were enrolled in the study. Men were significantly older, taller, heavier, had higher BMIs, lower fat percentages, higher lean body mass, and wider waist circumferences compared to women (*p* < 0.001). Among 185 women recruited, 132 were menopausal women with an average 10.17 (SD = 7.22) years since menopause. The majority of the subjects were from Hulu Langat district in Klang Valley. Most subjects had sedentary jobs (94.0%), a monthly income <RM 7640 (95.6%) and at least secondary school education (48.8%). Regarding dietary and lifestyle practices, overall, most subjects seldom consumed dairy products (68.9%) and calcium supplement (82.2%). Although most subjects were non-smokers (82.6%) and rarely drank alcohol (61.3%), they drank coffee or tea regularly (80.7%). Additionally, most of them were minimally active (43.3%). With regards to bone health, the percentage of normal, osteopenia and osteoporosis among Malaysians Chinese aged 40 years and above were 35.4%, 49.3%, and 15.3%, respectively (Table 1). 

### 3.2. Knowledge and Beliefs towards Osteoporosis

The mean total knowledge score of all subjects was 67.58% (SD = 12.99). Overall, the subjects scored higher in general knowledge (73.38%, SD = 17.08) than prevention knowledge regarding osteoporosis (61.76%, SD = 16.85). Most participants correctly identified the phrase “osteoporosis makes bones weaker, brittle, and more likely to break, causing fractures” as being “true” (96.2%), while most participants incorrectly identified the phrase “the regular intake of calcium supplements can lead to the formation of kidney stones” as being “true” (14.7%). Descriptive analysis for each item in the questionnaire in general and based on sexes can be found in Supplementary Material 1 (Appendix A).

The average total score for health beliefs among subjects (n = 367) was 71.93% (SD = 5.61). Overall, subjects had a low perception of susceptibility to osteoporosis because only 47.4% agreed or strongly agreed that they had high chances of getting osteoporosis. The perception of subjects on the seriousness of osteoporosis, benefits of exercise, and benefits of calcium intake was high. Subjects also perceived that they had low barriers to exercise and calcium intake. For calcium intake, 80.9% of the subjects agreed that they liked calcium-rich foods, and 83.7% agreed that calcium-rich food was not costly. Subjects also reported a high health motivation, as evidenced by their willingness to look for new information related to health (73.6%), do regular health check-up (68.1%) and follow the recommendation to keep them healthy (88.5%). Descriptive analysis for each item in the questionnaire in general and based on sexes can be found in Supplementary Material 2 (Appendix A). The level of knowledge and beliefs regarding bone health of the subjects was shown in Table 2.

### 3.3. Comparison of Socio-Demographic Factors on Knowledge and Beliefs Regarding Osteoporosis

In terms of osteoporosis knowledge, significant differences (*p* < 0.001) were found between sexes (Table 3). Women (76.57%) scored significantly higher in general knowledge regarding osteoporosis compared to men (70.14%) (*p* < 0.05). However, the score of knowledge regarding prevention of osteoporosis was similar between both sexes (*p* > 0.05). No significant differences in knowledge regarding osteoporosis based on other demographic characteristics were noted in this study (*p* > 0.05).

Beliefs regarding osteoporosis were associated with age, sex, education level, and parity. Comparison between sexes showed that women perceived significantly higher susceptibility to osteoporosis (60.37%) compared to men (56.59%) (*p* = 0.011). On the other hand, men (44.73%) had less barrier to exercise than women (52.54%) (*p* < 0.001). Subjects aged 61–70 years had significantly higher health motivation compared to other age groups (*p* = 0.045). Those with at least a university degree and above also had lesser barriers to exercise (*p* = 0.022) and higher health motivation (*p* = 0.005). Among women, those with 1 to 3 lifetime pregnancies had significantly higher health motivation compared to others (*p* = 0.021) (Table 3).

### 3.4. Comparison of Dietary and Lifestyle Practices with Socio-Demographic Factors

This study demonstrated that dietary and/or lifestyle practices were significantly different based on sex, nature of job, and education level. Sex differences were revealed in the dairy intake, calcium supplement intake, coffee or tea intake, alcohol drinking, and smoking status (*p* < 0.05). More women consumed dairy products (38.9% vs. 23.1% in men) and calcium supplements (23.2% vs. 12.1% in men) regularly than men. More men consumed coffee or tea (87.4% vs. 74.1% in women), alcohol (45.6% vs. 31.9% in women), and smoked cigarettes (30.8% vs. 4.3% in women) than women. More manual workers (63.6%) consumed dairy products than those with a sedentary job (29%). In terms of education, those with at least a university degree were less likely to be a smoker compared to others (*p* = 0.048) (Table 4).

### 3.5. Correlation between Osteoporosis Knowledge, Osteoporosis Health Beliefs, Dietary and Lifestyle Practices, and Bone Health Status

Knowledge regarding osteoporosis was correlated with several aspects of osteoprotective beliefs and practices. Osteoporosis knowledge was positively correlated with perceived benefits of exercise (r = 0.205, *p* ≤ 0.001), perceived benefits of calcium intake (r = 0.183, *p* ≤ 0.001), and health motivation (r = 0.138, *p* = 0.008). A negative relationship was noted between osteoporosis knowledge with barriers to exercise (r = −0.121, *p* = 0.020) and barriers to calcium intake (r = −0.208, *p* ≤ 0.001) A positive relationship between coffee or tea intake and osteoporosis knowledge was noted (r = 0.123, *p* = 0.019). No other significant interrelationships between knowledge regarding osteoporosis, dietary, and lifestyle practices were found (*p* < 0.05). Health beliefs regarding osteoporosis correlated significantly with several dietary and lifestyle choices of the subjects. Perceived susceptibility to osteoporosis correlated significantly and positively with intakes of dairy (r = 0.120, *p* = 0.022) and calcium supplements (r = 0.113, *p* = 0.030). Additionally, health motivation correlated positively with dairy intake (r = 0.170, *p* ≤ 0.001). Low barrier to exercise (r = −0.227, *p* ≤ 0.001) and high health motivation (r = 0.124, *p* = 0.018) also predicted high physical activity status. Health beliefs were not associated with other aspects of lifestyle practices (*p* > 0.05). Knowledge and beliefs regarding osteoporosis were not associated with the bone health status of the subjects (*p* < 0.05) (Table 5).

## 4. Discussion

This study revealed that Malaysian Chinese had moderate knowledge and a high level of health beliefs regarding osteoporosis, whereby the subjects recognized their susceptibility to osteoporosis, were aware of the seriousness of osteoporosis, and the benefits of prevention or preventive activity, such as calcium intake and exercise. However, the knowledge and beliefs did not translate into good dietary and lifestyle practices. Furthermore, knowledge regarding osteoporosis was associated with the sex of the subjects while health beliefs were associated with age, sex, education level, and parity. Additionally, dietary and lifestyle practices were associated with sex, nature of job, and education level of the subjects. The present study also found that knowledge regarding osteoporosis was correlated positively with health beliefs (benefits of exercise, benefits of calcium intake and health motivation domains) and coffee or tea intake but not with other lifestyle practices. Health beliefs regarding osteoporosis was associated positively with physical activity, dairy and calcium intake of the subjects. However, the relationship between bone health with knowledge, beliefs and practices regarding osteoporosis was not significant in this study.

This study indicated a moderate level of knowledge regarding osteoporosis among subjects with an average total score of 73.38 (17.08). This finding is in line with previous studies on knowledge regarding osteoporosis among Malaysian adult populations [10,13]; adult women (aged 25–65 years old) in India [34]; and adult women in New Zealand women (aged 20–49 years old) [35]. Specifically, 59.1% of the subjects were unaware that osteoporosis does not cause knee pain, presumably because the subjects conflate the effects of osteoarthritis with osteoporosis. This misconception may be easily addressed in future osteoporosis prevention campaign. Of the subjects, 81.7% also indicated that regular intake of calcium supplements can lead to the formation of kidney stones. This could be a barrier for them to consume calcium supplements. Additionally, 59.1% subjects did not know that glucocorticoids could increase the risk of osteoporosis probably because they are not familiar with glucocorticoids and their side effects. Sub-analysis based on sex showed that women had a significantly better general knowledge regarding osteoporosis as compared to men. Other studies comparing the knowledge of women with that of men also reported similar findings [36,37]. It is suggested that women are at greater risk to develop osteoporosis; thus, they have higher osteoporosis awareness compared to men.

Perception of personal susceptibility and seriousness of disease are important in influencing behavioral change in disease prevention program [35]. In the present study, we noted that women perceived a higher susceptibility to osteoporosis compared to men. This shows that Malaysian women are aware that they are at risk of osteoporosis, probably through public campaigns and commercial advertisement of dairy products. Similar findings related to sex differences in susceptibility awareness to osteoporosis have been documented in other population-based studies [38]. The barrier to exercise was found to be influenced by age, sex and education level. Men, middle-aged subjects and subjects with at least a university degree had less barrier to exercise. Previous studies involving subjects ≥50 years in the United States [36], Chinese aged ≥40 years in Northwest China [39] and subjects from Turkey aged ≥35 years [38] also showed that men have lower barriers to exercise compared to women. The reasons for this difference could be due to gender stereotyping of Asian women to avoid vigorous and outdoor activities. There are certain reasons for not participating in outdoor activities among Asian women, including modesty or avoidance of mixed-sex activity and fear of going out alone [40]. On the other hand, it was not surprising that middle-aged subjects had less barriers to exercise, because they were more energetic and physically fit compared to the elderly. Additionally, higher health motivation was demonstrated among those with at least a university degree and women with 1–3 lifetime pregnancies. We postulated that these populations are more concerned about their health due to education awareness and reduction in bone mass related to parity [41], although this relationship is not straightforward.

Dietary (dairy products, calcium supplements, coffee or tea) and lifestyle practices (smoking, alcohol, physical activity) among subjects were also examined in this study. Despite a moderate level of knowledge and a high level of health beliefs regarding osteoporosis, osteoprotective practices among subjects were poor. Osteoporosis preventive practices were not performed regularly by the subjects. Only 31.1% consumed dairy products and 17.7% consumed calcium supplements. The Malaysian Dietary Guidelines (2017) recommend adequate calcium from low-fat milk and its products and other calcium-rich food sources to prevent osteoporosis [42]. Calcium intake is particularly important in postmenopausal and elderly women because calcium deficiency is associated with reduced bone density. Low calcium intake has been reported to be one of the risk factors for osteoporosis amongst Asian women [43], even though supplementation (i.e., calcium and vitamin D) may reduce the rate of bone loss and fracture in adults [44]. The low dairy and calcium intake of Chinese subjects in this study might be related to the prevalent perception that calcium causes renal stones, the high prevalence of lactose intolerance among Malaysian Chinese [45], the absence of dairy products as ingredients in Chinese dishes, aversion of the taste [46], and concerns about weight gain and the fat/cholesterol content of some calcium-rich foods [47]. Further studies are needed to understand the barriers for them to practice osteoprotective behaviors. Optimal physical activities, especially weight-bearing exercises, can increase bone mass and reduce osteoporosis risk [44]. However, most subjects in this study were minimally active. This might be related to the presence of other comorbidities, the lack of facilities and the warm and humid weather typical of Malaysia. Nevertheless, the actual reasons for lacking exercise among the subjects may require further investigation. Furthermore, most subjects in this study were non-smokers and non-alcohol drinkers. Long-term cigarette smoking and consuming excessive amounts of alcohol might compromise bone mineral density, and their relationships were identified in several epidemiological studies [48,49,50]. Many subjects (80.7%) were regular coffee or tea drinkers. The relationship between coffee and tea intake and bone health is still debatable because some studies suggest that moderate consumption might be beneficial to bone health [51,52], and only high consumption might increase the risk of osteoporosis [53,54].

This study also illustrated that the knowledge score was positively associated with health beliefs regarding osteoporosis. According to various health promotion models, people with a positive health beliefs and adequate knowledge will be more likely to engage in health-related behaviors. The association between knowledge and practices regarding osteoporosis was mostly not significant in this study. Similar findings were obtained in a study involving women of different life stages, in which no significant associations among knowledge scores, total calcium intake and weight-bearing physical activity level were found [47]. The lack of relationship might imply that the subjects associated these practices with health benefits other than osteoprotection. On the other hand, a significant relationship between health beliefs and osteoprotective practices was detected in this study, particularly between perceived susceptibility to osteoporosis and health motivation with dairy/calcium intake, barriers to exercise and health motivation to physical activity. This shows that health beliefs are more important in driving individuals to adopt osteoprotective behaviors than knowledge alone. No significant correlation was found between subjects’ bone health and osteoporosis knowledge, health beliefs and practices regarding osteoporosis within this study. This may imply that bone health among Malaysians Chinese aged 40 years and above might be influenced by other extrinsic or intrinsic factors, which require further investigation.

The present study should be interpreted within the context of its strength and limitations. We only determine the intake of dairy products and calcium supplements, but not other Chinese food sources that are rich in calcium. As the study design was cross-sectional, conclusions on the long-term influence of knowledge, beliefs, and practices on bone health could not be drawn. We excluded individuals with prior fractures from participating in this study, thus may have introduced selection bias in this study, whereby the subjects recruited were healthier than the general population. We did not adopt the full version of OPAAT and OHBS to evaluate the knowledge and beliefs of the subjects because subjects complained of fatigue during the pilot study. Hence, the researchers selected the most relevant questionnaires and retested them in the pilot study. To the best of our knowledge, this was the first study which attempted to determine the association between knowledge, beliefs, and practices regarding osteoporosis and bone health status among middle-aged and elderly Chinese Malaysians of both sexes using DXA.

## 5. Conclusions

In summary, the present data suggest that Malaysians Chinese aged 40 years and above have moderate knowledge and high level of beliefs regarding osteoporosis. However, these did not translate into good dietary and lifestyle practices. Women had higher general knowledge and perceived susceptibility to osteoporosis, but men had fewer barriers to exercise; however, women consumed calcium and dairy products more regularly than men. Health beliefs had a stronger relationship to several behaviors related to osteoporosis than knowledge. No correlation was found between knowledge, beliefs, and practices regarding osteoporosis with bone health determined by DXA in this study. Since osteoporosis knowledge, health beliefs, and preventive practices are modifiable, they should be targeted in future intervention programs to prevent osteoporosis. Men should also be targeted in future osteoporosis awareness campaign considering that they have poorer knowledge and practices (especially dairy products and calcium supplement intake) regarding bone health than women.

## Figures and Tables

**Table 1 ijerph-16-01787-t001:** Characteristics of subjects.

Variable of Interest	Mean (SD)		
	Men (n = 192)	Women (n = 185)	Overall (n = 367)
**Age (years)**	60.75 (9.03)	57.56 (8.62)	59.14 (8.96)
**Age of menarche (years)**	-	13.12 (1.79)	-
**Number of children (n)**		2.11 (1.38)	
**Age of menopause (years)**		51.31 (3.44), n = 132	
**Years since menopause (years)**		10.17 (7.22), n = 132	
**Body anthropometry**			
Height (m)	167.67 (5.82)	155.59 (5.37)	161.58 (8.23)
Weight	68.13 (8.43)	55.59 (8.82)	61.81 (10.66)
BMI (kg/m^2^)	24.25 (2.97)	22.95 (3.39)	23.59 (3.25)
Body fat percentage (%)	28.39 (4.53)	37.60 (4.73)	33.03 (6.53)
Lean body mass	46.06 (4.87)	32.72 (4.04)	39.33 (8.04)
Waist circumference (cm)	86.76 (10.20)	78.26 (10.47)	82.47 (11.17)
	**n (%)**		
**Age range**			
Middle age (40–59 years old)	77 (42.3)	107 (57.8)	184 (50.1)
Elderly (60 years old and above)	105 (57.7)	78 (42.2)	183 (49.9)
**District**			
Klang	8 (4.4)	5 (2.7)	13 (3.5)
Hulu Langat	132 (72.5)	160 (86.5)	292 (79.6)
Hulu Selangor	2 (1.1)	-	2 (0.5)
Petaling	36 (19.8)	18 (9.7)	54 (14.7)
Gombak	4 (2.2)	2 (1.1)	6 (1.6)
**Marital status**			
Single	10 (5.5)	23 (12.4)	33 (9.0)
Married	172 (94.5)	162 (87.6)	334 (91.0)
**Nature of job**			
Manual	11 (6.0)	11 (5.9)	22 (6.0)
Sedentary	171 (94.0)	174 (94.1)	345 (94.0)
**Classification of monthly income ^$^**			
B40	168 (92.3)	183 (98.9)	351 (95.6)
M40	14 (7.7)	2 (1.1)	16 (4.4)
**Highest education level**			
No formal education	1 (0.5)	3 (1.6)	4 (1.1)
Primary school	20 (11.0)	15 (8.1)	35 (9.5)
Secondary school	74 (40.7)	105 (56.8)	179 (48.8)
Certificate/diploma	52 (28.6)	37 (20.0)	89 (24.3)
University degree	26 (14.3)	18 (9.7)	44 (12.0)
Postgraduate	9 (4.9)	7 (3.8)	16 (4.4)
**Current menstrual status**			
Pre-menopause	-	31 (16.8)	-
Peri-menopause		22 (11.9)	
Menopause		132 (71.4)	
**Number of lifetime pregnancies (parity)**			
Nulliparous	-	34 (18.4)	-
1–3 Pregnancies		99 (53.5)	
More than 3 Pregnancies		52 (28.1)	
**Dairy intake**			
Do not drink	140 (76.9)	113 (61.1)	253 (68.9)
Regular drinker	42 (23.1)	72 (38.9)	114 (31.1)
**Calcium supplement intake**			
Yes	22 (12.1)	43 (23.2)	65 (17.7)
No	160 (87.9)	142 (76.8)	302 (82.2)
**Coffee or tea intake**			
Do not drink	23 (12.6)	48 (25.9)	71 (19.3)
Regular drinker	159 (87.4)	137 (74.1)	296 (80.7)
**Alcohol drinking**			
Non-drinker	99 (54.4)	126 (68.1)	225 (61.3)
Ever-drinker	83 (45.6)	59 (31.9)	142 (38.7)
**Smoking status**			
Non-smoker	126 (69.2)	177 (95.7)	303 (82.6)
Ever-smoker	56 (30.8)	8 (4.3)	64 (17.4)
**Physical activity status**			
Inactive	68 (37.4)	63 (34.1)	131 (35.7)
Minimally-Active	72 (29.6)	87 (47.0)	159 (43.3)
HEPA Active	42 (231)	35 (18.9)	77 (21.0)
**Body mass index**			
Normal	101 (55.5)	109 (58.9)	210 (57.2)
Underweight	19 (10.4)	34 (18.4)	53 (14.4)
Overweight	62 (34.1)	42 (22.7)	104 (28.3)
**Bone health status**			
Normal	77 (42.3)	53 (28.6)	130 (35.4)
Osteopenia	84 (46.2)	97 (52.4)	181 (49.3)
Osteoporosis	21 (11.5)	35 (18.9)	56 (15.3)

SD: Standard deviation. ^$^ B40, subjects with household income < RM 7640; M40, subjects with household income RM 7640–15,159.

**Table 2 ijerph-16-01787-t002:** Level of knowledge and beliefs towards osteoporosis.

Aspects	Overall (n = 367), n (%)	Mean (%) (SD)
**General Knowledge regarding Osteoporosis (Q1–6)**		
Low (0–50%)	61 (16.6)	73.38 (17.08)
Moderate (51–69%)	112 (30.5)	
High (70–100%)	194 (52.9)	
**Knowledge regarding Prevention of Osteoporosis (Q7–12)**		
Low (0–50%)	143 (39.0)	61.76 (16.85)
Moderate (51–69%)	147 (40.1)	
High (70–100%)	77 (21.0)	
**Total Knowledge regarding Osteoporosis (Q1–12)**		
Low (0–50%)	58 (15.8)	67.58 (12.99)
Moderate (51–69%)	148 (40.3)	
High (70–100%)	161 (43.9)	
**I: Perceived Susceptibility to Osteoporosis (Q1-2)**		
Low (0–50%)	145 (39.5)	58.50 (14.34)
Moderate (51–69%)	118 (32.2)	
High (70–100%)	104 (28.3)	
**II: Perceived Seriousness of Osteoporosis (Q3)**		
Low (0–50%)	66 (18.0)	72.75 (18.55)
Moderate (51–69%)	47 (12.8)	
High (70–100%)	254 (69.2)	
**III: Perceived Benefits of Exercise (Q4)**		
Low (0–50%)	18 (4.9)	80.49 (12.84)
Moderate (51–69%)	14 (3.8)	
High (70–100%)	335 (91.3)	
**IV: Perceived Benefits of Calcium Intake (Q5)**		
Low (0–50%)	31 (8.4)	75.80 (13.66)
Moderate (51–69%)	39 (10.6)	
High (70–100%)	297 (80.9)	
**V: Barriers to exercise (Q6–7)**		
Low (0–50%)	252 (68.7)	48.67 (14.68)
Moderate (51–69%)	60 (16.3)	
High (70–100%)	55 (15.0)	
**VI: Barriers to calcium intake (Q8–9)**		
Low (0–50%)	330 (89.9)	43.79 (8.27)
Moderate (51–69%)	27 (7.4)	
High (70–100%)	10 (2.7)	
**VII: Health motivation (Q10–12)**		
Low (0–50%)	4 (1.1)	73.99 (9.43)
Moderate (51–69%)	107 (29.2)	
High (70–100%)	256 (69.8)	
**Total Beliefs regarding Osteoporosis (Q1–12)**		
Low (0–50%)	-	71.93 (5.61)
Moderate (51–69%)	108 (29.4)	
High (70–100%)	259 (0.6)	

**Table 3 ijerph-16-01787-t003:** Comparison of osteoporosis knowledge and beliefs with socio-demographic factors.

Variable	Categories	N	Mean (%) (SD)										
			Knowledge			Health Beliefs							
			General	Prevention	Total	I	II	III	IV	V	VI	VII	I-VII
**Age Range (Years)**	40–59	71	74.09 (15.94)	61.05 (16.21)	67.57 (12.22)	58.53 (14.84)	72.39 (18.90)	80.54 (12.96)	74.89 (14.41)	69.62 (15.62)	76.85 (7.81)	73.51 (8.94)	71.53 (5.63)
	60 and above	125	72.68 (18.16)	62.48 (17.50)	67.58 (13.75)	72.39 (18.90)	73.11 (18.23)	80.44 (12.75)	76.72 (12.85)	73.06 (13.48)	75.57 (8.68)	74.46 (9.90)	72.32 (5.58)
	***p*-value**		**0.428**	**0.418**	**0.997**	**0.967**	**0.709**	**0.937**	**0.200**	**0.025 ^a^**	**0.140**	**0.337**	**0.177**
**Sex**	Men	182	70.14 (18.79)	62.09 (16.54)	66.12 (13.92)	56.59 (14.96)	73.52 (18.80)	81.54 (12.87)	75.71 (13.84)	44.73 (13.85)	44.12 (8.73)	74.07 (9.08)	72.37 (5.40)
	Women	185	76.57 (14.57)	61.44 (17.19)	69.01 (11.87)	60.37 (13.49)	72.00 (18.32)	79.46 (12.76)	75.89 (13.53)	52.54 (14.47)	43.46 (7.80)	73.91 (9.79)	71.49 (5.79)
	***p*-value**		**≤0.001** **^a^**	**0.714**	**0.033** **^a^**	**0.011 ^a^**	**0.434**	**0.121**	**0.901**	**≤0.001 ^a^**	**0.444**	**0.874**	**0.131**
**Marital Status**	Single	33	75.25 (15.66)	66.16 (17.42)	70.71 (11.99)	58.79 (17.99)	73.33 (17.80)	80.00 (15.00)	76.97 (11.32)	51.21 (15.36)	42.73 (7.61)	73.13 (9.20)	71.62 (4.99)
	Married	334	73.20 (17.22)	61.33 (16.76)	67.27 (13.06)	58.47 (13.96)	72.69 (18.65)	80.54 (12.63)	75.69 (13.88)	48.41 (14.61)	43.89 (8.34)	74.07 (9.46)	71.96 (5.68)
	***p-*value**		**0.481**	**0.135**	**0.147**	**0.923**	**0.846**	**0.843**	**0.547**	**0.322**	**0.411**	**0.579**	**0.716**
**Nature of Job**	Sedentary	345	73.38 (17.11)	62.17 (16.95)	67.78 (12.99)	58.29 (14.41)	72.58 (18.69)	80.64 (12.88)	75.83 (13.60)	48.55 (14.83)	43.65 (8.03)	74.03 (9.18)	71.94 (5.55)
	Manual	22	73.48 (16.79)	55.30 (13.98)	64.39 (12.90)	61.82 (12.96)	75.45 (16.25)	78.18 (12.20)	75.45 (15.03)	50.45 (12.14)	45.91 (11.41)	73.33 (12.01)	71.67 (6.71)
	***p*-value**		**0.978**	**0.064**	**0.245**	**0.231**	**0.433**	**0.371**	**0.911**	**0.489**	**0.215**	**0.807**	**0.852**
**Classification of Monthly Income ^$^**	B40	351	73.46 (17.11)	61.73 (16.88)	67.59 (12.98)	58.80 (14.19)	72.88 (18.53)	80.28 (12.96)	75.90 (13.40)	48.83 (14.64)	43.82 (8.40)	74.00 (9.51)	71.95 (5.62)
	M40	16	71.87 (16.91)	62.50 (16.66)	67.19 (13.77)	51.88 (16.42)	70.00 (19.32)	85.00 (8.94)	73.75 (18.93)	45.00 (15.49)	43.13 (4.79)	73.75 (7.88)	71.46 (5.57)
	***p*-value**		**0.718**	**0.858**	**0.903**	**0.059**	**0.545**	**0.151**	**0.660**	**0.308**	**0.744**	**0.919**	**0.734**
**Highest Education Level**	No formal education and primary school	39	69.66 (18.68)	60.68 (18.53)	65.17 (14.92)	53.59 (11.11)	70.77 (20.44)	77.44 (14.64)	75.38 (12.53)	48.97 (15.18)	46.41 (10.88)	70.94 (12.56)	69.40 (6.09)
	Secondary school	179	72.44 (17.42)	61.27 (17.00)	66.85 (12.86)	59.78 (13.57)	73.30 (18.48)	79.78 (12.18)	76.76 (13.09)	50.89 (14.93)	43.69 (8.13)	73.11 (8.85)	71.63 (5.34)
	Certificate/diploma	89	74.16 (15.07)	62.55 (17.28)	68.35 (12.64)	57.64 (16.92)	71.91 (18.76)	81.35 (13.07)	75.51 (13.40)	46.29 (14.88)	43.71 (6.81)	75.13 (7.96)	72.45 (5.23)
	University degree and above	60	77.50 (17.31)	62.78 (14.83)	70.14 (12.40)	59.17 (13.81)	73.67 (17.46)	83.33 (12.84)	73.67 (16.26)	45.33 (12.28)	42.50 (8.56)	76.89 (10.01)	73.67 (6.04)
	***p*-value**		**0.108**	**0.868**	**0.209**	**0.092**	**0.820**	**0.106**	**0.491**	**0.022 ^b^**	**0.144**	**0.005 ^b^**	**0.002 ^b^**
**Number of Lifetime Pregnancies (Parity)**	Nulliparous	34	79.41 (12.35)	62.75 (16.44)	71.08 (10.71)	60.29 (16.23)	71.18 (17.88)	80.00 (17.75)	75.88 (13.73)	52.35 (13.72)	43.53 (8.49)	74.51 (9.91)	71.61 (5.36)
	1–3 Pregnancies	99	74.41 (15.49)	61.78 (17.86)	68.10 (12.77)	61.31 (12.51)	71.92 (19.15)	79.60 (12.44)	76.97 (13.51)	53.33 (15.19)	43.33 (8.08)	75.35 (9.35)	71.99 (6.19)
	More than 3 Pregnancies	52	78.85 (13.65)	59.93 (16.59)	69.39 (10.79)	58.65 (13.43)	72.69 (17.28)	78.85 (9.21)	73.85 (13.45)	51.15 (13.67)	43.65 (6.87)	70.77 (9.98)	70.44 (5.23)
	***p*-value**		**0.093**	**0.731**	**0.436**	**0.518**	**0.931**	**0.909**	**0.405**	**0.679**	**0.970**	**0.021^b^**	**0.299**
**Current Menstrual Status**	Pre-menopause	31	74.19 (14.81)	60.22 (20.04)	67.20 (13.77)	60.97 (15.13)	74.19 (20.78)	79.35 (15.04)	74.84 (17.10)	56.77 (15.58)	40.97 (5.97)	73.76 (9.26)	71.34 (6.42)
	Peri-menopause	22	78.03 (13.98)	62.88 (13.54)	70.46 (10.20)	60.00 (15.11)	70.00 (20.24)	82.73 (9.35)	75.45 (13.71)	52.27 (15.10)	43.64 (6.58)	72.12 (10.00)	71.06 (6.25)
	Menopause	132	76.89 (14.65)	61.49 (17.12)	69.19 (11.69)	60.30 (12.90)	71.81 (17.47)	78.94 (12.68)	76.21 (12.63)	51.59 (14.02)	44.02 (8.28)	74.24 (9.91)	71.60 (5.60)
	***p*-value**		**0.576**	**0.857**	**0.587**	**0.961**	**0.700**	**0.437**	**0.869**	**0.199**	**0.146**	**0.642**	**0.915**

Low: 0–50%; Moderate 51–69%; High: 70% and above; a: indicates significant difference of *p* < 0.05 as assessed using independent t-test; b: indicates significant difference of *p* < 0.05 as assessed using post hoc analysis of ANOVA among the group in the same column. ^$^ B40, subjects with household income < RM 7640; M40, subjects with household income RM 7640–15,159.

**Table 4 ijerph-16-01787-t004:** Comparison of dietary and lifestyle practices with socio-demographic factors.

Variable	Categories	N	n (%)												
			Dairy Intake		Calcium Supplement Intake		Coffee or Tea Intake		Alcohol Drinking		Smoking Status		Physical Activity Status		
			Do Not Drink	Regularly	Yes	No	Do Not Drink	Regularly	Non-Drinker	Ever-Drinker	Non-Smoker	Ever-Smoker	Inactive	Minimally-Active	HEPA-Active
**Age Range (Years)**	40–59	184	122 (66.3)	62 (33.7)	34 (18.5)	150 (81.5)	40 (21.7)	144 (78.3)	107 (58.2)	77 (41.8)	148 (80.4)	36 (19.6)	67 (36.4)	80 (43.5)	37 (20.1)
	60 and above	183	131 (71.6)	52 (28.4)	31 (16.9)	152 (83.1)	31 (16.9)	152 (83.1)	118 (64.5)	65 (35.5)	155 (84.7)	28 (15.3)	64 (35.0)	79 (43.2)	40 (21.9)
	***p*-value**		**0.276**		**0.700**		**0.246**		**0.214**		**0.283**		**0.680**		
**Sex**	Men	182	140 (76.9)	42 (23.1)	22 (12.1)	160 (87.9)	23 (12.6)	159 (87.4)	99 (54.4)	83 (45.6)	126 (69.2)	56 (30.8)	68 (37.4)	72 (39.6)	42 (23.1)
	Women	185	113 (61.1)	72 (38.9)	43 (23.2)	142 (76.8)	48 (25.9)	137 (74.1)	126 (68.1)	59 (31.9)	177 (95.7)	8 (4.3)	63 (34.1)	87 (47.0)	35 (18.9)
	***p*-value**		**0.001 ^a^**		**0.005 ^a^**		**0.001 ^a^**		**0.007 ^a^**		**≤0.001 ^a^**		**0.330**		
**Marital Status**	Single	33	20 (60.6)	13 (39.4)	4 (12.1)	29 (87.9)	4 (12.1)	29 (87.9)	21 (63.6)	12 (36.4)	28 (84.8)	5 (15.2)	17 (51.5)	13 (39.4)	3 (9.1)
	Married	334	233 (69.8)	101 (30.2)	61 (18.3)	273 (81.7)	67 (20.1)	267 (79.9)	204 (61.1)	130 (38.9)	275 (82.3)	59 (17.7)	114 (34.1)	146 (43.7)	74 (22.2)
	***p*-value**		**0.278**		**0.378**		**0.271**		**0.773**		**0.717**		**0.077**		
**Nature of Job**	Sedentary	345	245 (71.0)	100 (29.0)	63 (18.3)	282 (81.7)	66 (19.1)	279 (80.9)	212 (61.4)	133 (38.6)	286 (82.9)	59 (17.1)	120 (34.8)	154 (44.6)	71 (20.6)
	Manual	22	8 (36.4)	14 (63.6)	2 (9.1)	20 (90.9)	5 (22.7)	17 (77.3)	13 (59.1)	9 (40.9)	17 (77.3)	5 (22.7)	11 (50.0)	5 (22.7)	6 (27.3)
	***p*-value**		**≤0.001 ^a^**		**0.275**		**0.679**		**0.826**		**0.500**		**0.130**		
**Classification of Monthly Income ^$^**	B40	351	241 (68.7)	110 (31.3)	61 (17.4)	290 (82.6)	69 (19.7)	282 (80.3)	215 (61.3)	136 (38.7)	291 (82.9)	60 (17.1)	126 (35.9)	153 (43.6)	72 (20.5)
	M40	16	12 (75.0)	4 (25.0)	4 (25.0)	12 (75.0)	2 (12.5)	14 (87.5)	10 (62.5)	6 (37.5)	12 (75.0)	4 (25.0)	5 (31.3)	6 (37.5)	5 (31.3)
	***p*-value**		**0.592**		**0.435**		**0.478**		**0.920**		**0.415**		**0.587**		
**Highest Education Level**	No formal education and Primary school	39	28 (71.8)	11 (28.2)	8 (20.5)	31 (79.5)	7 (17.9)	32 (82.1)	18 (46.2)	21 (53.8)	30 (76.9)	9 (23.1)	15 (38.5)	14 (35.9)	10 (25.6)
	Secondary school	179	124 (69.3)	55 (30.7)	32 (17.9)	147 (82.1)	40 (22.3)	139 (77.7)	113 (63.1)	66 (36.9)	140 (78.2)	39 (21.8)	63 (35.2)	75 (41.9)	41 (22.9)
	Certificate/diploma	89	58 (65.2)	31 (34.8)	13 (14.6)	76 (85.4)	12 (13.5)	77 (71.8)	57 (64.0)	32 (36.0)	79 (88.8)	10 (11.2)	30 (33.7)	43 (48.3)	16 (18.0)
	University degree and above	60	43 (71.7)	17 (28.3)	12 (20.0)	48 (80.0)	12 (20.0)	48 (80.0)	37 (61.7)	23 (38.3)	54 (90.0)	6 (10.0)	23 (38.3)	27 (45.0)	10 (16.7)
	***p*-value**		**0.812**		**0.797**		**0.383**		**0.383**		**0.048^b^**		**0.799**		
**Number of Lifetime Pregnancies (Parity)**	Nulliparous	34	21 (61.8)	13 (38.2)	6 (17.6)	28 (82.4)	5 (14.7)	29 (85.3)	22 (64.7)	12 (35.3)	30 (88.2)	4 (11.8)	14 (41.2)	17 (50.0)	3 (8.8)
	1–3 Pregnancies	99	59 (59.6)	40 (40.4)	25 (25.3)	74 (74.7)	28 (28.3)	71 (71.7)	69 (69.7)	30 (30.3)	96 (97.0)	3 (3.0)	36 (36.4)	43 (43.4)	20 (20.2)
	More than 3 Pregnancies	52	33 (63.5)	19 (36.5)	12 (23.1)	40 (76.9)	15 (28.8)	37 (71.2)	35 (67.3)	17 (32.7)	51 (98.1)	1 (12.5)	13 (25.0)	27 (51.9)	12 (23.1)
	***p*-value**		**0.895**		**0.663**		**0.254**		**0.856**		**0.059**		**0.297**		
**Current Menstrual Status**	Pre-menopause	31	17 (54.8)	14 (45.2)	8 (25.8)	23 (74.2)	10 (32.3)	21 (67.7)	18 (58.1)	13 (41.9)	29 (93.5)	2 (6.5)	11 (35.5)	16 (51.6)	4 (12.9)
	Peri-menopause	22	13 (59.1)	9 (40.9)	5 (22.7)	17 (77.3)	7 (31.8)	15 (68.2)	14 (63.6)	8 (36.4)	20 (90.9)	2 (9.1)	7 (31.8)	9 (40.9)	6 (27.3)
	Menopause	132	83 (62.9)	49 (37.1)	30 (22.7)	102 (77.3)	31 (23.5)	101 (76.5)	94 (71.2)	38 (28.8)	128 (97.0)	4 (3.0)	45 (34.1)	62 (47.0)	25 (18.9)
	***p*-value**		**0.696**		**0.934**		**0.483**		**0.328**		**0.353**		**0.778**		

a: indicates significant difference of *p* < 0.05 as assessed using independent t-test; b: indicated significant difference of *p* < 0.05 as assessed using post hoc analysis of ANOVA among the group in the same column. ^$^ B40, subjects with household income < RM 7640; M40, subjects with household income RM 7640–15,159.

**Table 5 ijerph-16-01787-t005:** Correlation between osteoporosis knowledge, health beliefs, dietary and lifestyle practices, and bone health status (n = 367).

		Overall (n = 367)														
		A	B	C	D	E	F	G	H	I	J	K	L	M	N	O
**A. Total knowledge score**	**r**	-	0.018	0.022	**0.205**	**0.183**	**−0.121**	**−0.208**	**0.138**	0.059	0.100	**0.123**	0.049	−0.092	0.073	0.016
	**P**		0.726	0.678	**≤0.001**	**≤0.001**	**0.020**	**≤0.001**	**0.008**	0.260	0.054	**0.019**	0.353	0.078	0.162	0.767
**B. Perceived Susceptibility to Osteoporosis**	**r**		-	**0.210**	−0.029	0.018	**0.171**	0.023	**0.141**	**0.120**	**0.113**	0.050	−0.007	−0.087	−0.078	0.069
	**P**			**≤0.001**	0.584	0.731	**0.001**	0.665	**0.007**	**0.022**	**0.030**	0.340	0.898	0.095	0.138	0.185
**C. Perceived Seriousness of Osteoporosis**	**r**			-	0.102	0.087	0.047	0.069	0.040	0.002	−0.019	−0.013	−0.021	−0.053	0.034	−0.004
	**P**				0.050	0.097	0.373	0.187	0.449	0.970	0.719	0.806	0.683	0.313	0.521	0.946
**D. Perceived Benefits of Exercise**	**r**				-	**0.149**	**−0.179**	**−0.146**	**0.136**	0.066	−0.029	0.040	0.057	−0.062	0.082	−0.057
	**P**					**0.004**	**0.001**	**0.005**	**0.009**	0.206	0.581	0.442	0.277	0.233	0.115	−0.274
**E. Perceived Benefits of Calcium Intake**	**r**					-	−0.058	−0.023	**0.146**	0.086	0.038	0.182	0.064	0.015	0.069	−0.026
	**P**						0.268	0.655	**0.005**	0.102	0.467	0.117	0.222	0.774	0.190	0.613
**F. Barriers to Exercise**	**r**						-	**0.107**	**−0.133**	−0.079	−0.006	−0.045	−0.065	−0.012	**−0.227**	−0.038
	**P**							**0.040**	**0.011**	0.129	0.902	0.394	0.214	0.818	**≤0.001**	0.470
**G. Barriers to Calcium Intake**	**r**							-	−0.067	−0.008	−0.074	−0.043	0.015	0.024	0.020	0.082
	**P**								0.201	0.873	0.154	0.415	0.774	0.647	0.704	0.116
**H. Health Motivation**	**r**								-	**0.170**	0.008	0.039	0.032	−0.037	**0.124**	−0.019
	**P**									**0.001**	0.876	0.458	0.542	0.480	**0.018**	0.716
**I. Dairy Intake**	**r**									-	−0.018	0.060	0.071	**−0.122**	−0.134	−0.017
	**P**										0.726	0.248	0.173	**0.019**	0.010 *	0.740
**J. Calcium Supplement Intake**	**r**										-	−0.062	0.042	−0.082	−0.014	0.095
	**P**											0.237	0.425	0.119	0.791	0.069
**K. Coffee or Tea Intake**	r											-	**0.120**	**0.152**	**0.108**	−0.064
	**P**												**0.022**	**0.003**	**0.039**	0.223
**L. Alcohol Drinking**	**r**												-	**0.210**	**0.090**	−0.052
	**P**													**≤0.001**	**0.085**	0.319
**M. Smoking Status**	**r**													-	−0.006	−0.054
	**P**														0.914	0.306
**N. Physical Activity Status**	**r**														-	0.006
	**P**															0.909
**O. Bone Health Status**	**r**															-
	**P**															

Number in bold indicated significant association between the compared variables.

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
