# Peer review of "Knowledge, Beliefs, Dietary, and Lifestyle Practices Related to Bone Health among Middle-Aged and Elderly Chinese in Klang Valley, Malaysia"

_ijerph, 2019, doi:10.3390/ijerph16101787_

Round 1
Reviewer 1 Report
This article conducted a cross-sectional survey of the relationship between osteoporosis and bone health knowledge, beliefs and practice in Malaysian Chinese aged 40 and over for up to 9 months. It helps us to understand the knowledge of osteoporosis in the population, and to have a certain significance for the future promotion of osteoporosis knowledge and the development of interventions. The research adopts the questionnaire method. The content of the questionnaire is comprehensive, and it is objective to use the multiple-choice question. However, this study is a cross-sectional study, the results of the study are yet to be further verified. In addition, this study excluded lower limb fractures 2 years before the screening date, or low-impact fractures after 50 years , or could not complete the questionnaire or screening procedure ,which may overestimate the subjects' knowledge of bone health.
Author Response
Thank you for the comments given. We have edited the manuscript according to the comments given. We have attached the response to reviewer at the PDF file below.

Reviewer 2 Report
Please see the attachment.

Author Response

(The authors gave the same response as above.)

Round 2
Reviewer 1 Report
My questions have been addressed.